# *Eriocheir sinensis feminization-1c* (*Fem-1c*) and Its Predicted miRNAs Involved in Sexual Development and Regulation

**DOI:** 10.3390/ani13111813

**Published:** 2023-05-30

**Authors:** Dandan Zhu, Tianyi Feng, Nan Mo, Rui Han, Wentao Lu, Zhaoxia Cui

**Affiliations:** 1School of Marine Sciences, Ningbo University, Ningbo 315020, China; zhudd2015@163.com (D.Z.);; 2Laboratory for Marine Biology and Biotechnology, Qingdao National Laboratory for Marine Science and Technology, Qingdao 266071, China; 3DECAPODA Biology Science and Technology Co., Ltd. (Lianyungang), Lianyungang 222000, China

**Keywords:** feminization-1c, intersex, sex differentiation, miRNA, *Eriocheir sinensis*

## Abstract

**Simple Summary:**

The *Fem-1c* gene in *Eriocheir sinensis* (Henri Milne Edwards, 1854) (*EsFem-1c*) is related to sex differentiation; however, its functional mechanism is less reported. In this study, qRT-PCR results of *EsFem-1c* from the intersex crab implied that *EsFem-1c* plays a role in crab androgenic gland (AG) development. RNAi and scanning electron microscope observations revealed that *EsFem-1c* influenced sexual development in *E. sinensis*. A dual-luciferase reporter assay demonstrated that tcf-miR-315-5p binding to the 3′UTR sequence effectively inhibited the translation of the *EsFem-1c* gene, and tcf-miR-307 binding to the alternative splicing region in the 3′UTR sequence could increase *EsFem-1c* expression. In summary, this study provides a better understanding of the molecular regulation mechanism of *EsFem-1c*.

**Abstract:**

*Feminization-1c* (*Fem-1c*) is important for sex differentiation in the model organism *Caenorhabditis elegans*. In our previous study, the basic molecular characteristics of the *Fem-1c* gene (*EsFem-1c*) in *Eriocheir sinensis* (Henri Milne Edwards, 1854) were cloned to determine the relationship with sex differentiation. In this study, the genomic sequence of *EsFem-1c* contained five exons and four introns, with an exceptionally long 3′UTR sequence. The qRT-PCR results of *EsFem-1c* demonstrated lower tissue expression in the androgenic gland of the intersex crab than the normal male crab, implying that *EsFem-1c* plays a role in crab AG development. RNA interference experiments and morphological observations of juvenile and mature crabs indicated that *EsFem-1c* influences sexual development in *E. sinensis*. A dual-luciferase reporter assay disclosed that tcf-miR-315-5p effectively inhibits the translation of the *EsFem-1c* gene, influencing male development. An intriguing finding was that miRNA tcf-miR-307 could increase *EsFem-1c* expression by binding to the alternative splicing region with a length of 248 bp (ASR-248) in the 3′UTR sequence. The present research contributes to a better understanding of the molecular regulation mechanism of *EsFem-1c* and provides a resource for future studies of the miRNA-mediated regulation of sexual development and regulation in *E. sinensis*.

## 1. Introduction

The *Feminization-1* (*Fem-1*) gene is a well-known key regulator to control male sexual development in the cascade sex determination pathway of *Caenorhabditis elegans*; together with the Cullin-2 protein, and by ubiquitin-mediated proteolysis, it directs the proteasome-mediated degradation of the Transformer-1 (TRA-1) protein [1,2,3,4]. Doniach (1984) and Kimble (1984) demonstrated that a *Fem-1* mutation in *C. elegans* can result in a sex switch from male to hermaphrodite [5,6]. *Fem-1* influences courtship and sex determination in *Drosophila melanogaster* [7]. Several mammalian and invertebrate species have been found to carry *Fem-1* family (*Fem-1a*, *-1b*, and *-1c*) genes [8,9,10]. They are distinguished by the conserved existence of ankyrin repeat domains, and are assumed to have different biological activities [11].

Recently, *Fem-1* genes in several crustaceans have been studied. It was observed that the sexually dimorphic expression pattern is related to the development of sexual phenotype and the formation of germ cells [12]. *Mrfem-1* and *Mnfem-1* genes have specific strong positive signals in the ovary of *Macrobrachium rosenbergii* and *Macrobrachium nipponense*, and *Mnfem-1* can interact with proteins containing ankyrin, cathepsin L, insulinase or ubiquitin, indicating that *Mnfem-1* and *Mrfem-1* could function in prawn sexual development and ovarian development [13,14]. The *fem-1c* gene has also been identified in *Hyriopsis cumingii and E. sinensis* differently regulating gonadal differentiation development. The *HcFem-1c* gene in *Hyriopsis cumingii* maintains testicular function and induces sperm discharge [2]. *EsFem-1c* transcripts are abundant in muscle for *Eriocheir sinensis*, in addition to expression in gonads [15]. Although these findings provide some insights into *Fem-1c*’s role in crustacean sex development and differentiation, they fail to elucidate its mechanisms.

As known, 3′ untranslated regions (3′UTRs) of messenger RNAs (mRNAs) notoriously regulate mRNA-regulation processes by serving as hubs for post-transcriptional control, such as stability, localization, and translation [16]. Functional elements in the 3′UTR sequences, such as Musashi binding elements (MBEs) and K boxes, play important roles in gene regulation [17,18]. The 3′UTR is also well-known as a target binding site for microRNA (miRNA), which regulates mRNA expression after transcription [19]. After forming the miRNA-induced silencing complex (miRISC), miRISC binds to specific mRNA and activates the regulation function [20]. In addition, alternative cleavage and polyadenylation (APA) and natural antisense transcript (NAT) have been demonstrated to influence 3′UTRs functions [21]. APA is an RNA transcription regulation mechanism that allows mRNA with different 3′ ends to be generated from a single gene using polyadenylation (poly(A)) sites selectively, resulting in alternative transcript and protein isoforms [22,23,24]. The reverse strand of endogenous encoding or non-encoding molecules transcribes NAT [25]. An NAT of *Pvfem-1* is only present in the oocyte nucleus of the subadult female Pacific white shrimp, *Penaeus vannamei*, indicating a possible post-transcriptional mechanism in ovary development [26].

MiRNAs play a role in crustacean immunity, reproduction, cell proliferation, and apoptosis. Many studies on *E. sinensis* have explained how miRNAs target genes to control the post-transcriptional regulatory processes. For example, miR-2 and miR-133 can down-regulate cooperatively the 3′ UTR of the cyclin B gene in *E. sinensis* via cytoplasmic polyadenylation element (CPE) and miRNA-binding sites [27]. Moreover, miR-133b regulates the role of *SAP30* in the spermatogenesis of *E. sinensis* by regulating histone deacetylation levels to ensure normal spermatogenesis and the sperm decondensation nuclear mechanism [28]. Furthermore, during the *White Spot Syndrome Virus* (*WSSV*) challenge in *E. sinensis*, MiR-7 suppresses Myd88-ILF2-(IL-16L) signaling pathways and increases *WSSV* replication [29]. The miR-17-3p is highly regulated during spermatogonia differentiation and development in *E. sinensis* [30]. The miR-305-5p, miR-263a-5p, and miR-7-5p play major regulatory roles in the metamorphosis of *E. sinensis* [31]. In *E. sinensis*, miR-34 and let-7b can down-regulate insulin-like androgenic gland hormone (*IAG*) gene expression, whereas miR-9-5p, let-7, and miR-8915 can down-regulate Double-sex (*Dsx*) gene expression [32].

The Chinese mitten crab, *E. sinensis* (Henri Milne Edwards, 1854), is an economically important aquatic animal in China. In previous studies, *Fem-1* genes were discovered to play important roles in the early gonadal development of *E. sinensis* based on genomic and transcriptomic data [15]. In this study, we assessed the expression profiles of *EsFem-1c* in normal and intersex crabs (genetic female crabs with androgenic gland). RNA interference (RNAi) was used to investigate the effects of *EsFem-1c* on the gender characteristics of juvenile and mature crabs. The function of potential miRNAs binding to the 3′UTR of *EsFem-1c* was identified and analyzed by constructing the small RNAs (sRNAs) sequencing database. Finally, it was found that the gender-biased expression pattern of the alternative spliced 3′UTR with a length of 248 bp (ASR-248) in the *EsFem-1c* gene was the basis for regulating the function of *EsFem-1c*.

## 2. Materials and Methods

### 2.1. Collection of Animals and Samples

*E. sinensis* was collected from crab hatcheries in Jiangsu province, China. Because the number of experimental individuals and types of target tissue samples for different experiments differed, the crabs were kept conventionally in our laboratory for further experimentation and sampling.

After anesthetizing the mature normal crabs (body weight: 94.27 ± 8.14 g) and the intersex crabs (body weight: 91.32 ± 6.29 g) on ice for 5 min, different tissues including muscle, ovaries or testes, and androgenic glands were dissected and quickly frozen in liquid nitrogen for subsequent RNA extraction.

We selected female crabs for ovary sampling based on the description by Gu and He (1997), which were in stages Ⅱ, Ⅲ, and Ⅳ [33]. Testis samples were obtained from male crabs at various stages of development, including spermatocyte, spermatid, and sperm stages [34]. Furthermore, the androgenic gland tissues were sampled at the proliferation, synthesis, and secretion stages [35].

Male and female reproductive systems were collected to construct the miRNAs database. Ovary (O) and spermathecal (SA) samples were collected from female crabs. In contrast, testis (T), vas deferens (VD), accessory sex glands (ASG), ejaculatory ducts (ED), seminal vesicles (SV), and androgenic glands (AG) were sampled from male crabs.

All animal treatments and experimental procedures in the present study were carried out in accordance with Ningbo University’s Guide for the Use of Experimental Animals.

### 2.2. DNA, RNA, and miRNAs Preparation and cDNA Template Synthesis

Using the Marine Animal DNA kit (TIANGEN, Beijing, China), the genomic DNA of muscles was extracted from female and male crabs as directed. Total RNA was extracted from various samples using TRIZOL^®^ reagent (Invitrogen, Waltham, MA, USA) according to manufacturer’s instructions. The cDNA template was synthesized from 1 µg of total RNA using the PrimeScript™ RT reagent Kit (with gDNA Eraser) (TaKaRa, Shiga, Japan). Moreover, according to the protocol, miRNAs were extracted from the reproduction system of male and female crabs using the PureLink miRNA Isolation Kit (Invitrogen, USA). Then, 2 µg miRNA was used to synthesize the miRNA cDNA template using the miRNA First Strand cDNA Synthesis Kit (Vazyme, Nanjing, China).

### 2.3. Cloning and Analysis of Fem-1c 3′UTR Sequence from Eriocheir Sinensis

To determine the sequence accuracy of *EsFem-1c* 3′UTR for future studies, the polymerase chain reaction (PCR) template and specific primers were utilized to perform the PCR (Table 1). Primer Premier software (version 6.0, Premier Biosoft, San Francisco, CA, USA) was used to design the primers [36].

A 25.0 µL mixture of approximately 0.5 µL of each primer (10 mM), 0.5 µL *Ex Taq* polymerase (TaKaRa, Japan), 1.0 µL cDNA template, 2.5 µL 10 × *Ex Taq* polymerase buffer, 2.0 µL dNTP, and 18.0 µL double-distilled water was used for the PCR. The PCR program included pre-denaturation at 95 °C for 3 min, denaturation at 95 °C for 30 s, annealing at 58 °C for 30 s, and extension at 72 °C for 80 s for 35 cycles. The products separated by gel electrophoresis were sequenced by Youkang (Yongkang, China). The sequences were trimmed of vector contamination using the VecScreen tool (https://www.ncbi.nlm.nih.gov/tools/vecscreen/ (accessed on 1 April 2023)) [37].

Poly (A) signal Miner (http://dnafsminer.bic.nus.edu.sg/PolyA.html (accessed on 1 April 2023)) was utilized to detect the position of the poly (A) signal [38]. The elements in 3′UTRs were analyzed using UTRScan (http://itbtools.ba.itb.cnr.it/utrscan (accessed on 1 April 2023)) [39].

### 2.4. Synthesis of dsRNA for Silencing EsFem-1c Gene by RNA Interference In Vivo

Specific primers cloned and cleaved a 406 bp fragment of the *EsFem-1c* gene (GenBank: KR108012) and a fragment of the *EGFP* gene (GenBank: U55761) into the pGEM^@^-T Easy Vector (Promega Biotech, Beijing, China) (Table 1). After sequencing, double-stranded RNAs (dsRNAs) including dsRNA *EGFP* and dsRNA *EsFem-1c* were synthesized using the TranscriptAidTM T7 high-yield transcription kit following the manufacturer’s protocol (Thermo Scientific Inc., San Jose, CA, USA). Nanodrop2000 (Thermo Scientific Inc., USA) was used to measure the concentration.

### 2.5. Target-Silencing EsFem-1c Gene in Juvenile and Mature Crabs

In this experiment, dsRNA *EsFem-1c* was injected into the mature mitten crabs (body weight 94.27 ± 8.14 g) and juvenile Ⅰ crabs (average body weight 10 mg).

There were three treatment groups in mature crabs: the blank control group (saline solution), the negative control group (dsRNA *EGFP*), and the treated group (dsRNA *EsFem-1c*). In mature crabs, the injection dose was 1 µg/g body weight. The dsRNA was injected into the coelom through the arthrodial membrane between the third and fourth pleopod by a 100 μL micro syringe (Shanghai Anting, Shanghai, China). After 24 h, mature crabs were sampled for qRT-PCR, including the eyestalk ganglion and ovary in females and testis and AG in male crabs. To analyze the molecular function of dsRNA *EsFem-1c*, *IAG* (GenBank ID: KU724192.1) and crustacean female sex hormone 1 (*CFSH-1*) (GenBank ID: OP351640) genes of *E. sinensis* were detected [40].

The blank control group (BC) and the treated group (dsRNA *EsFem-1c*) were conducted on juvenile Ⅰ crabs at 1 µg/g of body weight. The injection position was also at the arthrodial membrane between the third and fourth pleopod using the IM11-2 and M-152 micro injection system (NARISHIGE, Tokyo, Japan). Approximately 15 days after, the whole body was sampled at the juvenile Ⅲ stage and held in 1× phosphate-buffered saline (1× PBS) solution and 4% paraformaldehyde.

### 2.6. The Morphological Observation of Juvenile Crab

The general anatomical structures of the juvenile crab were observed under the stereomicroscope (Olympus, Tokyo, Japan). Scanning electron microscope (SEM) (Hitachi, Tokyo, Japan) was used to obtain micrographs of juvenile crabs. The juvenile crabs’ samples were prepared by the standard process described by Zhu et al. [40]. The samples for observation were prepared through ethanol gradient dehydration and tert-butyl alcohol replacement. After being frozen for 20 h at −20 °C, the samples were dried for 24 h in the Martin Christ Alpha 1—4LD plus freeze dryer (Martin Christ, Osterode am Harz, Germany) before being sprayed with gold by a Hitachi E-1010 ion sputtering device (Hitachi, Japan). The length of male penis was obtained by the software Image J and converted according to the scale of SEM images.

### 2.7. Small RNA Libraries Construction and Sequencing

RNA (1 µg) was used to isolate sRNA and construct libraries using the Illumina TruSeq small RNA sample preparation Kit (Illumina, San Diego, CA, USA). The sRNA was linked to the 5′ and 3′ adapters before being reverse-transcribed into cDNA. The cDNA library was evaluated using an Agilent Bioanalyzer 2100 system (Agilent Technologies, Santa Clara, CA, USA) after PCR amplification. Finally, the sequencing was performed via the Illumina Hiseq 2500 platform (LC-BIO, Hangzhou, China).

### 2.8. Bioinformatics Analysis of Sequencing Reads

After removing the low-quality reads, reads with 5′ adapter and ambiguous nucleotides, and reads without 3′ adapter or inserted tag from raw data using ACGT101-miR software (LC Sciences, Houston, TX, USA), the clean reads with lengths of 18–26 nt were processed for subsequent bioinformatics analyses. The selected clean reads were filtered in Rfam, Repbase, and other common RNA family databases, before being mapped to the genome of the mitten crab [41] and then to the miRBase 22.0 database [42] to identify novel or known miRNAs. The secondary structures, minimum free energy (*mfe*) value, and base bias of all identified miRNA on the first or each position were analyzed.

The miRNA read counts were normalized to norm values to measure miRNA expression [43]. Differential expression miRNAs (DEMs) among male and female reproduction were indicated as *p*-value and Log2 (Fold change). The threshold for significant differential expression was defined as *p*-value < 0.05. The miRanda (v3.3a) [44] and TargetScan (v5.0) [45] were employed to predict the target genes of the identified miRNAs and clarify their functions. Furthermore, the Gene Ontology (GO) enrichment analysis [46] was used to determine the biological functions of the predicted target genes. Moreover, the Kyoto Encyclopedia of Genes and Genomes (KEGG) pathway analysis was used to identify metabolic or signal transduction pathways [47].

### 2.9. Quantitative Real-Time PCR

The specific primers were used for quantitative real-time PCR (qRT-PCR) in the ABI 7500 system (Applied Biosystems, Waltham, MA, USA) with reverse transcription cDNA products of mRNA and miRNA (Table 2). A standard curve was used to measure amplification efficiency and ascertain primer specificity. Each sample was repeated in triplicates.

The qRT-PCR for mRNA was used to assay the transcript expression profiles of target genes in different cDNA templates using the TB Green Premix kit (2×) (TaKaRa, Japan). To standardize the mRNA expression, *Es-β-actin* (GenBank ID: ATO74508.1) was used as the reference gene. The reaction mixture and cycling conditions have been described in our previous studies [40]. EsFem-1c-qF/R had an amplification efficiency of 102.253%. EsCFSH-1-qF/R had an amplification efficiency of 105.895%. EsIAG-qF/R had an amplification efficiency of 101.054%, and the amplification efficiency of Es-β-actin-qF/R was 100.451%.

The qRT-PCR was used to determine the expression of miRNAs using the 2 × *Taq* Pro Universal SYBR qPCR Master Mix Kit (Vazyme, China). A 20.0 µL PCR mixture containing 10.0 µL of 2 × *Taq* Mix, 2.0 µL of diluted cDNA, 0.4 µL of miRNA specific primer, 0.4 µL of Universal Reverse Q primer, and 7.2 µL of RNase-free water was detected under the following conditions: denaturation at 95 °C for 10 s, annealing and extension at 60 °C for 30 s and 72 °C for 15 s, 40 cycles, respectively. To standardize miRNA expression, a single stranded U6 primer was used as an internal control [32].

### 2.10. Identification and Analysis of miRNAs Targeting EsFem-1c Gene

The binding sites located in the 3′UTR sequence of the *EsFem-1c* gene were detected using miRanda [44], RNA22 [48], and RNAHybird [49] based on the miRNA target prediction results. The analogs of miRNAs were synthesized chemically by General Biol. (General Biol., China). The present study used these analogs of predicted miRNAs to further investigate their effectiveness.

### 2.11. Luciferase 3′UTR Reporter Assay

To determine whether miRNAs can regulate the *EsFem-1c* gene, the interactions between them were examined using the dual-luciferase reporter assay in vitro.

The pGL4.10 [*luc2*] vector (Promega, China) was optimized to become the pGL4.10 [*luc2*]-basic plasmid (pGL-B), and the *Xba I* restriction site was inserted into the downstream of the firefly luciferase coding region via PCR amplification using specific primers (Table 3). Following enzyme digestion and purification, the 3′UTR sequence of *EsFem-1c* was cloned into the pGL-B to construct the pGL4.10 [*luc2*]-Fem1c-wild (Fem1c-WT). Similarly, the Mut Express^®^ Ⅱ Fast Mutagenesis Kit (Vazyme, China) and the specific primers were used to construct the pGL4.10 [*luc2*]-Fem1c-mutant (Fem1c-MUT) (Table 3).

The cell line HEK 293T was obtained from the Chinese Academy of Science cell bank (Shanghai, China) and cultured in Dulbecco’s Modified Eagle Medium (DMEM; Hyclone, Logan, UT, USA) supplemented with 1% 100 × Penicillin-streptomycin-neomycin solution (100 × PSN; Gibco, Shanghai, China), and 10% fetal bovine serum (FBS; Thermo Fisher Scientific, China) at 37 °C and 5% CO_2_ [50]. Cells were seeded into the 24-well plates at 1.5 × 10^5^ cells per well. When the cells were approximately 70–80% confluent, the co-transfection was conducted with 30 ng miRNAs mimics or the negative control (NC), 40 ng pGL4.74 [*hRluc/TK*] plasmid, and 200 ng Fem1c-WT or Fem1c-MUT plasmid, using Lipofectamine 3000 reagent (Thermo Fisher Scientific, Shanghai, China) according to the manufacturer’s protocol. Each sample was transfected in triplicate. Cells were incubated in serum- and antibiotic-free DMEM for 6 h, then 48 h transfection.

After harvesting cell lysates, luciferase activity was detected using the Dual-Luciferase Reporter Assay System kit (Promega, Beijing, China). The luc2/hRluc signal ratio was used to normalize the transfection efficiency between samples and calculate the relative luciferase measurement. The experiments were repeated thrice.

### 2.12. Statistical Analysis

Using the 2^−ΔΔCt^ method, the relative expression levels of target genes were calculated [51]. One-way Analysis of Variance (One-Way ANOVA) was performed using SPSS (version 23.0, Armonk, NY, USA). Differences were considered statistically significant at *p* < 0.05 [52].

## 3. Results

### 3.1. The Bioinformatics Analysis of EsFem-1c mRNA Sequence

The whole sequence of *EsFem-1c* was submitted to the NCBI database (GenBank number: KR108012) after being obtained from the *E. sinensis* genomic database [15]. There were five exons and four introns in the *EsFem-1c* sequence (Figure 1). The coding region measured 5976 base pairs (bp) and included a 1929 bp open read frame (ORF). The start codon was preceded by a 5′UTR of 205 bp, and a 3′UTR of 3914 bp followed the termination codon. The N-terminal region contained six consecutive ankyrin repeats (ANKs), while the C-terminal region contained two ANKs. The bioinformatics software forecast analysis results revealed one potential poly-adenylation signal (PAS; AATAAA) located in the 19 bp upstream of the poly (A) tract, and the analysis by the UTRScan software identified three K-boxes motifs (XTGTGATX) and six Musashi binding element sites (MBE; G/AT1-3AGT).

### 3.2. Expression Patterns Analysis of the EsFem-1c in Various Tissues of the Normal and Intersex Crabs

*EsFem-1c* gene expression was found in normal and intersex crab tissues, such as muscle, ovary, testes, and AG (Figure 2). *EsFem-1c* expression patterns in intersex crabs differed from those in normal female and male crabs. The gender of intersex crabs used in our experiment was distinguished to be female by our specific primers.

Notably, *EsFem-1c* expression was significantly higher in intersex crab muscles (*p* < 0.05), and the expression in normal male crabs was significantly higher than in normal female crabs (*p* < 0.05). Nevertheless, little difference existed between the normal female ovary and ovary-like tissue in intersex crabs. They were significantly higher (*p* < 0.05) than normal male testis. Regarding AG, *EsFem-1c* expression was significantly lower in intersex crabs than normal male crabs (*p* < 0.05).

### 3.3. The Expression Profiles of Sex Genes and Sexual Characteristics Change after Targeted-Silencing EsFem-1c in E. sinensis

Using dsRNA *EsFem-1c*, we investigated the roles of *EsFem-1c* in mature crabs and juveniles.

After being injected with dsRNA *EsFem-1c*, the crabs’ phenotypes, especially gonopore’s cover in the female crab and the AG in the male crab, changed from juvenile Ⅰ to Ⅲ, and the abnormal cleft structure of the gonopore’s cover was observed in female crabs at juvenile Ⅲ (Figure 3a). The appearance of the male penis did not change significantly when developed to juvenile Ⅲ (Figure 3b); however, the length became significantly longer than the control group (*p* < 0.05) (Figure 3c).

After target-silencing *EsFem-1c*, the expression levels of *EsIAG* and *EsCFSH-1* in mature crabs differed from the control group. In female crab EGs and ovaries, the expression of *EsCFSH-1* decreased significantly, with *EsFem-1c* silencing efficiencies of 65.1% and 80.2% (*p* < 0.05), respectively (Figure 3d,e). The silencing efficiency of dsRNA *EsFem-1c* in male crabs was 71.6% and 89.7% in AG and testis, respectively, while *EsIAG* in AGs and testes demonstrated an increased expression (*p* < 0.05) (Figure 3f,g).

### 3.4. The Expression of EsFem-1c in the Reproduction Tissues

EsFem-1c mRNA expression was detected in the ovary, testis, and AG at various developmental stages. According to qRT-PCR results (Figure 4), *EsFem-1c* expression decreased with ovarian development from Stage Ⅱ to Ⅳ (*p* < 0.05). The mRNA transcript of EsFem-1c from the spermatocyte stage of the testis and proliferation stage of AG revealed a downward trend (*p* < 0.05) (Figure 4). Interestingly, the peak level of *EsFem-1c* in gonadal developmental stages all appeared during the early stages of development (Figure 4).

### 3.5. Overview of Small RNAs Sequencing in the Reproduction System of E. sinensis

The six libraries (ES_F1, ES_F2, ES_F3, ES_M1, ES_M2, and ES_M3) were constructed using the *E. sinensis* reproduction system (Figure 5a). After removing the reads with the 3ADT and length filter and junk reads, 9,576,534, 7,908,841, 8,466,316, 11,347,130, 12,064,581, and 8,643,956 clean reads were obtained, which were then compared and filtered using databases (mRNA, RFam, and Repbase) to obtain sRNAs (Appendix A). The length distribution of sRNAs indicated similar trends in females and males, with the peak at 22 nt representing a typical characteristic of miRNAs (Figure 5b). By searching the genome of *E. sinensis* using the miRBase 22.0 and miRDeep2 programs, 154 known miRNAs and 611 novel miRNAs were identified. Nucleotide bias analysis revealed that uridine (U) was dominant at the first position of the known miRNAs, and U and guanosine (G) were predominant at each position of the reads (Appendix A).

Based on differential expression analysis of miRNAs, the male and female reproduction tissues of *E. sinensis* expressed 43 significantly DE miRNAs (DEMs) (*p* < 0.05) (Appendix A). Among these, 20 miRNAs presented significant up-regulation (female > male), while 23 miRNAs indicated down-regulation (male > female). PC-5p-16060 was the most significantly up-regulated miRNAs and PC-3p-74 was the most significantly down-regulated miRNAs. DEMs results were also used to generate a volcano plot depicting differential expression levels (Appendix A), and cluster analysis was used to estimate miRNA expression cluster profiles in different groups (Appendix A). As displayed in Figure 6, the qRT-PCR assay demonstrated that the results of the 16 miRNAs agreed with the norm values from the miRNA libraries.

The genes predicted to be targets of miRNAs were further classified using GO and KEGG pathway enrichment annotation analyses to investigate the potential pathways that these miRNAs might regulate. After analyzing the GO annotation results, target genes were identified to have the highest cornucopian gene counts in the deoxyribonucleotide biosynthetic process term (GO: 0009263) in the biological process, the integral component of the plasma membrane (GO: 0005887) in the cellular component, and the molecular function (GO: 0003674) in the molecular function (Appendix A). Moreover, KEGG pathway enrichment analysis depicted that target genes statistically focused on three significant pathways, including the circadian rhythm (map04710), insulin resistance (map04931), and longevity regulating pathway (map04211) (Appendix A).

### 3.6. The Interaction between the Identified miRNAs and EsFem-1c Gene by Dual-Luciferase Reporter Assay

After analysis by using the miRanda, RNA22, and RNAHybird programs, eight miRNAs were predicted to bind to the 3′UTR of *EsFem-1c* (Figure 7a). The miRNAs including tcf-let-7-3p (A), bmo-mir-6497-p5 (B), tcf-miR-315-5p (C), tcf-miR-7 (D), and tcf-miR-281-3p (E) were found near the termination codon in the 3′UTR sequence. PC-3p-120711 (F), tcf-let-7-5p (G), and tca-bantam-3p (H) were identified near the poly (A) site. The lowest free energy values were observed in bmo-mir-6497-p5 and tcf-let-7-5p (Appendix A). In terms of DEMs, female crabs had significantly higher expression of tcf-let-7-3p, bmo-mir-6497-p5, and PC-3p-120711, whereas males had higher expression of tcf-miR-315-5p and tcf-let-7-5p. There was no sex bias expression of tcf-miR-7, tca-bantam-3p, and tcf-miR-281-3p.

To determine whether these eight miRNAs could regulate *EsFem-1c*, the relative luciferase activity of these miRNAs was measured in vitro using the dual-luciferase reporter assay. While the tcf-miR-7 mimic did not significantly influence luciferase activity, the tcf-let-7-3p, bmo-mir-6497-p5, and tcf-miR-315-5p mimics resulted in an approximate 23%, 26%, and 32% decrease in luciferase activity compared to the NC group (Figure 7b).

The miRNA tcf-miR-315-5p was selected for further analysis and verification, because it significantly reduced luciferase activity. The GC% of tcf-miR-315-5p was 46.70%, and the *mfe* value was -21.30 kcal/mol (Figure 7c). The analysis of norm values and relative expression revealed that males had significantly higher levels of tcf-miR-315-5p than females (*p* < 0.05) (Figure 7d). As illustrated in Figure 7e, the results of co-transfecting displayed that the fluorescence intensity of cells treated with Fem1c-WT plasmid and tcf-miR-315-5p mimic was significantly decreased compared with that of the controls (*p* < 0.05), indicating that tcf-miR-315-5p inhibited the expression of the *EsFem-1c* gene by targeting its 3′UTR. Co-transfecting with Fem1c-MUT plasmid and tcf-miR-315-5p mimic also significantly decreased the luciferase activity (*p* < 0.05). However, the luciferase activity in the Fem1c-MUT plasmid group was higher than that in the Fem1c-WT plasmid group (*p* < 0.05) (Figure 7e).

### 3.7. The Analysis of the Alternative Splicing Region in 3′UTR of EsFem-1c

To validate the accuracy of the 3′UTR of *EsFem-1c* sequence, PCR amplification was performed in the muscle of 48 crabs (female: male = 1:1) with specific primers. There were three types of products: single band at 991 bp (a), single band at 743 bp (b), and double bands at 991 bp and 743 bp (ab). Figure 8b indicates that the ratio of a: ab: b in females differed from that in males. The female was 1:11:12, and the male was 3:9:12, with the female having a low single band at 991 bp. Sanger sequencing of these products was performed to confirm the findings, revealing that ASR-248 existed upstream of the poly (A) site, where tcf-miR-307 was observed (Figure 8a). The *mfe* of tcf-miR-307 was −26.0 kcal/mol, and the expression pattern revealed no sex bias (Figure 8c,d).

Co-transfection of the Fem1c-WT vector and tcf-miR-307 mimics significantly increased the relative activity of luciferase compared to the NC group (*p* < 0.05), as revealed in Figure 8e. The luciferase activity of the Fem1c-MUT vector and tcf-miR-307 mimics was significantly lower than that of the NC group (*p* < 0.05). Moreover, luciferase activity was lower in the Fem1c-MUT plasmid group than in the Fem1c-WT plasmid group (*p* < 0.05). These findings indicate that tcf-miR-307 could promote ASR-248 function.

## 4. Discussion

Several previous studies have demonstrated that *Fem-1c* has a broad expression abundance throughout the entire gonadal developmental process; for example, the expression level of *Lmfem-1c* gradually increases during testis development in *Locusta migratoria manilensis* [9]. The highest expression of *EsFem-1c* exists in the early development stage of gonads in *E. sinensis*, which is similar to that found in *Cherax quadricarinatus*, *Crassostrea gigas*, and *Penaeus vannamei* [26,53,54], indicating that *EsFem-1c* is involved in the development of oocyte and spermatogenesis. In contrast, tissue distribution analysis in the present study revealed that the expression level of the *EsFem-1c* gene in the normal ovary was higher than that in the normal testis, consistent with the result in *H. cumingii* [2], indicating that *EsFem-1c* plays a critical role in female gonads. Interestingly, *EsFem-1c* gene expression in intersex crab muscle is significantly higher than in male and female crabs. We can assume that the *EsFem-1c* gene has a regulatory pathway that allows excessive expression to move to other muscle tasks, such as molting or muscle growth and breakdown. Furthermore, *EsFem-1c* has a higher expression level in normal male AG than in intersex AG, whereas *MnFem-1b* is present in the AG of *M. nipponense* [12], indicating a potential role in male phenotype.

The length of the penis grows, and *EsIAG* expression increases in male mitten crabs after knocking down the *EsFem-1c*. IAG regulates male differentiation in decapods. A similar report indicates that the knocking down of *MrIAG* negatively regulates *MrFem-1* in the post-larvae of *M. rosenbergii* [55]. Based on the function of *EsIAG*, these findings suggest that *EsFem-1c* regulates male sexual development [56]. The present study also confirms the promoting effect of *EsFem-1c* on female sexual development by regulating *EsCFSH-1*, which has influenced the sexual characteristics of female crabs [40]. Therefore, *EsFem-1c* is thought to be an upstream regulator of *EsIAG* and *EsCFSH-1*, which regulate sex differentiation in *E. sinensis*.

GO enrichment and KEGG pathway analyses of the target genes predicted to bind to 43 DEMs revealed that they were significantly enriched in many pathways, including insulin, MAPK, Wnt, TGF, and the oxytocin signaling pathway, all of which have been associated with reproductive function [32]. For example, PC-3p-375 and bmo-miR-306a-5p were predicted to target the *glycogen synthase kinase 3* (*GSK3*) gene in the insulin signaling pathway. tcf-miR-10-5p and PC-3p-241774 targeted *C-jun-amino-terminal kinase-interacting protein 4* (*JIP-4*) and *neurofibromin* (*NF*). These genes were the key regulators of the MAPK pathway. The PC-3p-795 targeted Ras-related protein acted on the Wnt pathway. Our results agree with the analysis of miRNAs during larval metamorphic and gonadal development in *E. sinensis* [31].

Tcf-miR-315-5p expression is abundant in males, as in *Pteromalus puparum* [57], indicating that tcf-miR-315-5p may regulate male development. In addition, the miR-315 has been studied to act as one subset in *Drosophila melanogaster* sperm storage and the post-mating responses [58,59], as well as to regulate *Nilaparvata lugens* sex determination and differentiation [60]. The dual-luciferase reporter assay in the present study yielded a similar conclusion regarding the inhibition function of tcf-miR-315-5p on the *EsFem-1c* gene.

Furthermore, *EsFem-1c* expression decreased gradually with the development of the testis and AG. It can be speculated that tcf-miR-315-5p, as a key miRNA, is responsible for the repression of testis and AG development by inhibiting *EsFem-1c* expression. This is similar to the fact that miRNA let-7d and miR-34 are proven to control the testicular differentiation during gonadal development progress by inhibiting *EsIAG* gene expression [32].

ASR-248 identified in 3′UTR provides a new understanding of how 3′UTR influences the mRNA expression of *EsFem-1c* in *E. sinensis*. There are two theories regarding its function. The first can be used as the NAT. ASR-248 transcribes from 3′UTR, the non-protein-coding region. Considering the various roles of NAT, ASR-248 may influence the function of *EsFem-1c* by altering mRNA stability, masking miRNA-binding sites, and forming endogenous siRNAs [25,61]. An NAT observed in the *Pvfem-1* in *P. vannamei* suggests a negative post-transcriptional regulation to control female sexual trait differentiation [26]. The second possibility is APA. By forming an APA complex, APA can eventually produce several mRNA polyadenylation isoforms [24]. In this study, ASR-248 caused variations in the transcription products of *EsFem-1c*, which may influence the positive effect of miRNA tcf-miR-307. We can infer that different types of *EsFem-1c* transcripts may have different sex differentiation mechanisms, such as the dose effect of functional proteins. A similar mode has been reported on regulating *Dsx* gene expression in three insects, *D. melanogaster* [62], *Megaselia scalaris*, and *Anopheles gambiae* [63].

## 5. Conclusions

In conclusion, current research indicates that *EsFem-1c* is important in regulating sex differentiation and controlling sexual traits. The miRNA tcf-miR-315-5p functions as a suppressed regulator to influence the function of the 3′UTR of *EsFem-1c*. The miRNA tcf-miR-307 positively affects the *EsFem-1c* gene by binding to gender-biased ASR-248 in the 3′UTR. The findings of this study should help future research into the sex-related functional mechanism of *EsFem-1c* in *E. sinensis*.

## Figures and Tables

**Figure 1 animals-13-01813-f001:**
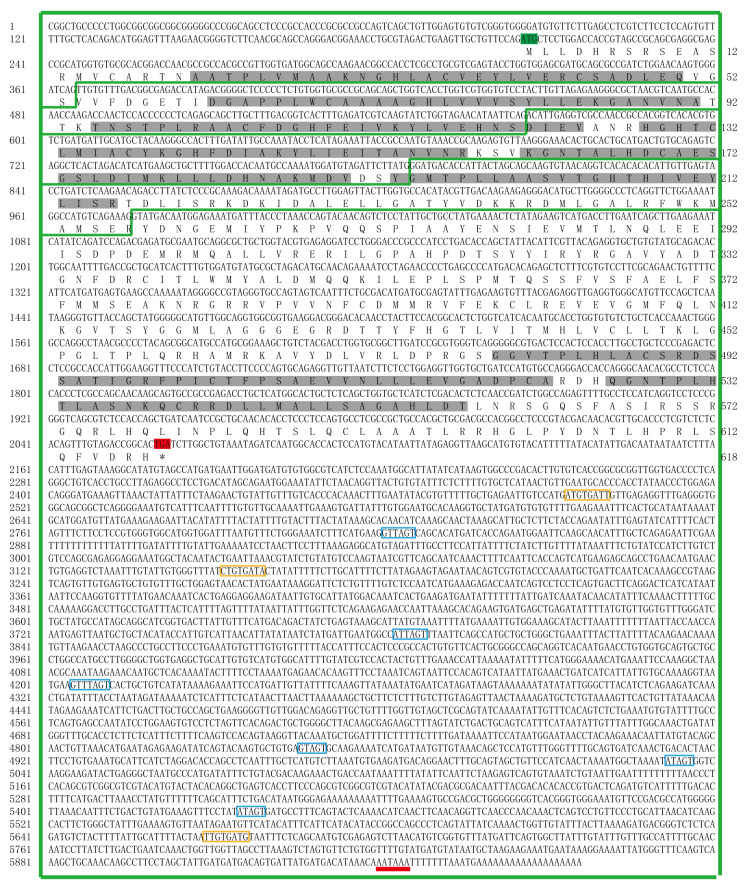
Nucleotide characteristics and deduced amino acid sequence of *EsFem-1c*. The *EsFem-1c* gene had five exons and four introns in its entire genomic structure. The green blanks enclose the five exons. The open reading frame (ORF) is the region between the initiation codon (highlighted in green) and the termination codon (highlighted in red, the amino acid is indicated by one asterisk), and the eight ANK motifs are highlighted in grey. The polyadenylation signal (PAS) is presented in red at the end of the 3′UTR. The K-box motif regions are enclosed in the yellow frame rectangle, while the Musashi binding element (MBE) motifs are in the blue frame rectangle.

**Figure 2 animals-13-01813-f002:**
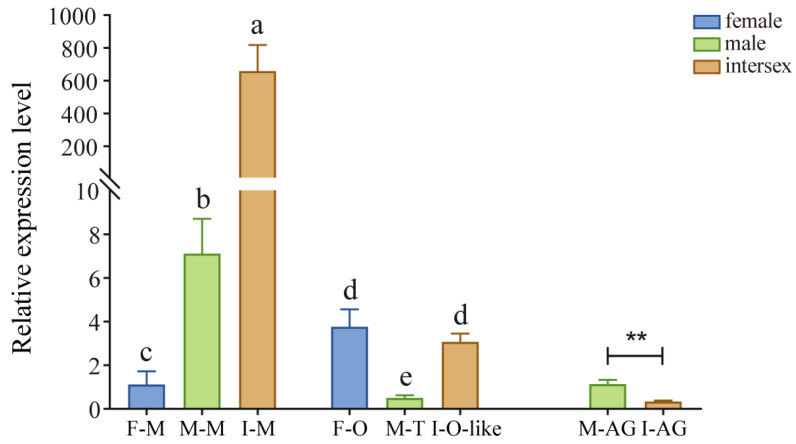
*EsFem-1c* tissue expression profiles in normal and intersex mitten crabs. This experiment used β-actin mRNA expression levels as an internal reference. F-M, M-M, and I-M represent the muscle of females, males, and intersex, respectively; M-AG and I-AG represent the androgenic gland of males and intersex, respectively; F-O represents normal ovary of females; M-T represents normal testis of males; I-O-like represents ovary-like tissue of intersex. Vertical bars represent the mean ± standard error of the mean (SEM) (*n* = 4). Asterisks and different lower-case letters indicated significant differences at *p* < 0.05.

**Figure 3 animals-13-01813-f003:**
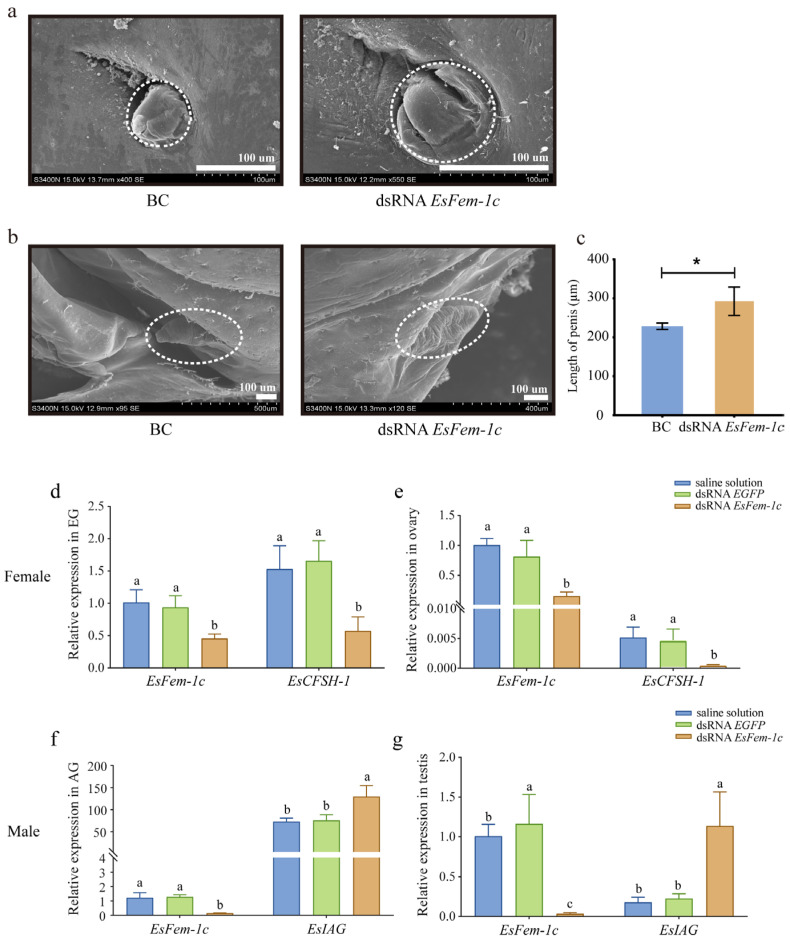
The analysis after injecting with the dsRNA *EsFem-1c*. (**a**,**b**) The altered characteristics of female gonopores and male penises in juvenile crabs. (**c**) A comparison of penis length in normal crabs treated with dsRNA *EsFem-1c* in juvenile Ⅲ stage. BC, blank control group. Data are presented as the mean ± SEM of separated individuals (*n* = 6). Asterisks indicate significant differences at *p* < 0.05. (**d**,**e**) Silencing efficiency of EsFem-1c and the expression changes of *EsCFSH-1* gene in female crab eyestalk ganglion (EG) and ovary. (**f**,**g**) Silencing efficiency of *EsFem-1c* and the expression changes of *EsIAG* gene in male crab androgenic gland (AG) and testis. Data are presented as the mean ± SEM of separated individuals (*n* = 6). For the expression of different genes in each group, different lower-case letters indicated significant differences at *p* < 0.05.

**Figure 4 animals-13-01813-f004:**
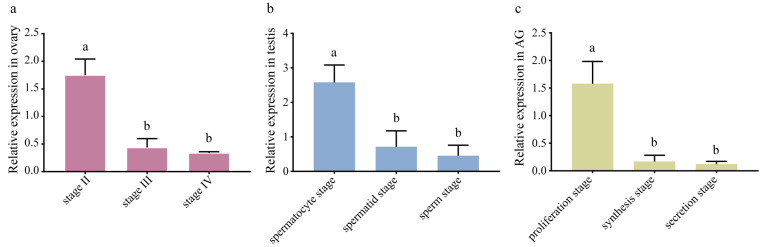
The expression of *EsFem-1c* in the reproduction tissues during developmental stages. (**a**) *EsFem-1c* expression level in the ovaries at stages Ⅱ, Ⅲ, and Ⅳ. (**b**) *EsFem-1c* expression level in the testis at the spermatocyte, spermatid, and sperm stages. (**c**) *EsFem-1c* expression level in AGs at the proliferation, synthesis, and secretion stages. Data are presented as the mean ± SEM of seperated individuals (*n* = 4). Statistical significance was accepted at *p* < 0.05 and indicated by different lower-case letters.

**Figure 5 animals-13-01813-f005:**
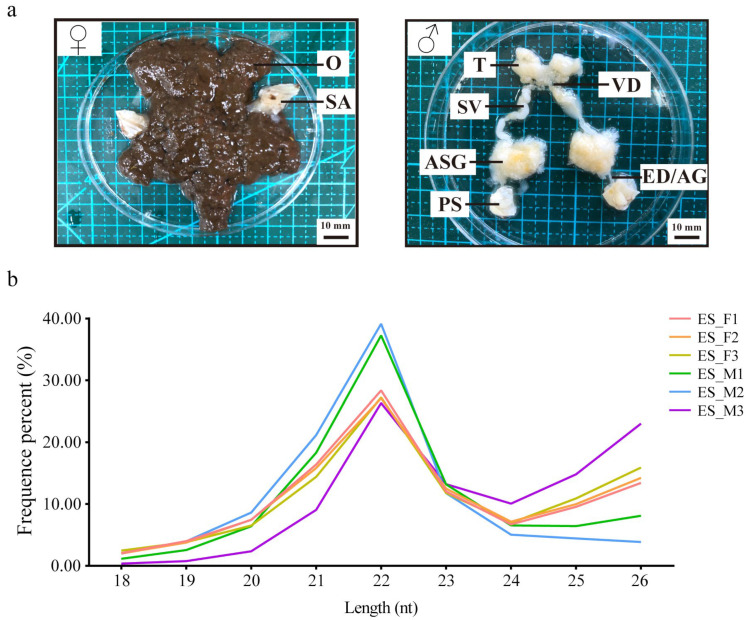
Basic information on six small RNA libraries of *E. sinensis*. (**a**) Female and male mitten crab reproduction systems. The female crab had ovary (O) and spermathecal (SA). The testis (T), vas deferens (VD), seminal vesicle (SV), accessory sex glands (ASG), ejaculatory duct (ED), androgenic gland (AG), and penis (PS) were all present in male crabs. (**b**) Sequence length distribution of six sRNA libraries. ES_F1, ES_F2, and ES_F3 were the three female libraries, while ES_M1, ES_M2, and ES_M3 were the three male libraries.

**Figure 6 animals-13-01813-f006:**
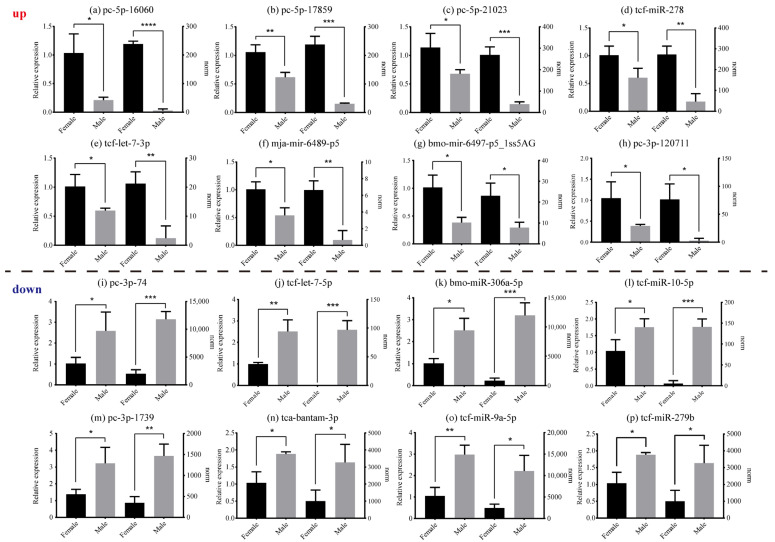
The relative expression level of 16 expressed miRNAs detected by qRT-PCR (**left**) and norm values derived from sequencing data (**right**). (**a**–**h**) The up-regulated group; the expression level of these eight miRNAs were higher in females than in males. (**i**–**p**) The down-regulated group; the expression level of these eight miRNAs were higher in males than in females. The norm values are normalized from miRNA read counts to count the miRNA expression. The values are presented as means ± SEM (*n* = 3). Statistical significance was accepted at *p* < 0.05 and marked by different asterisks.

**Figure 7 animals-13-01813-f007:**
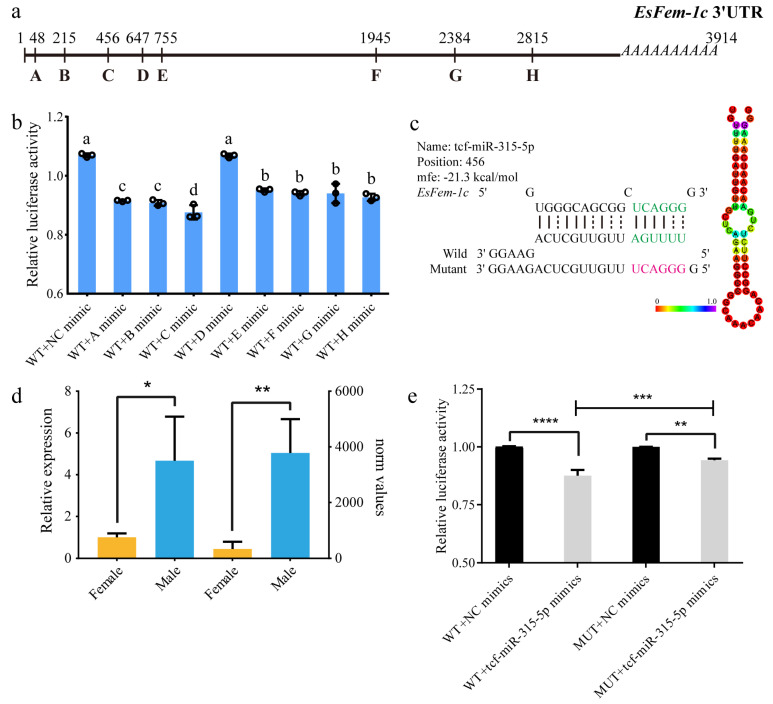
An analysis of the miRNAs predicted to target the 3′UTR of *EsFem-1c* gene. (**a**) Software-detected predicted binding position of eight miRNAs in the 3′UTR of *EsFem-1c*. (**b**) The relative luciferase activities of miRNAs tested by the dual-luciferase assay in HEK293T cells. A, tcf-let-7-3p; B, bmo-mir-6497-p5; C, tcf-miR-315-5p; D, tcf-miR-7; E, tcf-miR-281-3p; F, PC-3p-120711; G, tcf-let-7-5p; H, tca-bantam-3p. (**c**) The tcf-miR-315-5p miRNA sequence and secondary structure. (**d**) qRT-PCR analysis and norm values of the miRNA tcf-miR-315-5p revealed that this miRNA was highly expressed in males *E. sinensis*. (**e**) The relative luciferase activities in the wild and mutant assay. Values represent mean ± SEM of independent biological replicates (*n* = 3). The lower-case letters and asterisks indicate statistically significant differences at *p* < 0.05.

**Figure 8 animals-13-01813-f008:**
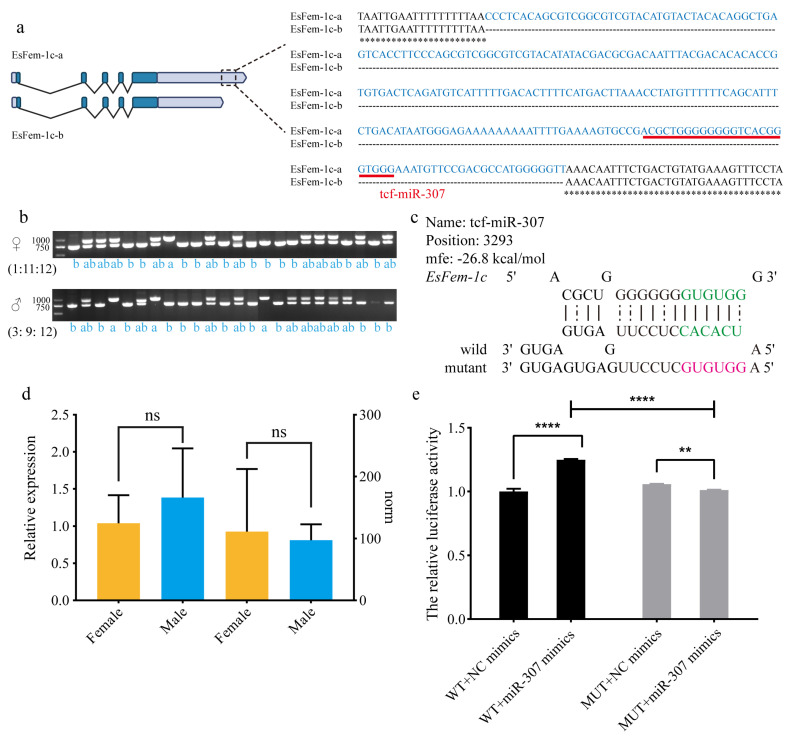
Analysis of the 248 bp region in the 3′UTR of *EsFem-1c*. (**a**) Software-detected the 248 bp sequence and binding sites of miRNA tcf-miR-307 in the 3′UTR of *EsFem-1c*. (**b**) PCR results of male and female crabs revealed three types of products: a single at 991 bp (a), a single at 743 bp (b), and a double at 991 bp and 743 bp (ab). (**c**) Information on the miRNA sequence and secondary structure of the miRNA tcf-miR-307. (**d**) qRT-PCR analysis and norm values of the tcf-miR-307 revealed that miRNA had no sex bias between the female and male *E. sinensis*. (**e**) The relative luciferase activities in the wild and mutant assay. The data are presented as mean ± SEM (*n* = 3). Statistical significance was accepted at *p* < 0.05 and indicated by different asterisks.

**Table 1 animals-13-01813-t001:** Primer sequences used to clone and synthesize dsRNA of target genes.

Primer	Sequence (5′–3′)	Applications
EsFem-1c-F1	CCAGTCATGAAGAGCAGCC	Gene cloning
EsFem-1c-R1	ATGGCCCAGAAGTGCTTG	
EsFem-1c-248-F2	CAGACTGCTGGGGCTTACAA	
EsFem-1c-248-R2	TTCATACATACCGGCCAGCC	
EsFem-1c-dsF	AGATCCAGACGAGATGCGAATG	Synthesis of dsRNA
EsFem-1c-dsR	CCGTCCTTCACCGCCAC	
EGFP-dsF	CACAAGTTCAGCGTGTCCG	
EGFP-dsR	AACCACTACCTGAGCACCCA	
Primer-T7	TAATACGACTCACTATAGGG	
Primer-SP6	ATTTAGGTGACACTATAG	

**Table 2 animals-13-01813-t002:** Primer sequences used for qRT-PCR.

Primer	Sequence (5′–3′)	Applications
EsFem-1c-qF	CTCAGTCCTGTTCCCTGCATT	qRT-PCR for mRNA
EsFem-1c-qR	AGGGCTGGCCGGTATGTAT	
Es-β-actin-qF	GCATCCACGAGACCACTTACA	
Es-β-actin-qR	CTCCTGCTTGCTGATCCACATC	
EsCFSH-1-qF	ATACGTTGAGCGCCAGATCC	
EsCFSH-1-qR	CAGAGCCACACATACGGAGC	
EsIAG-qF	GCTCCTACAAGCAGCACCC	
EsIAG-qR	AGGGTCTTCCAGATGGATCG	
tca-bantam-3p	TGAGATCATTGTGAAAGCTGATT	qRT-PCR for miRNAs
tcf-miR-281-3p	CTGTCATGGAGTTGCTCTCTTT	
tcf-miR-307	TCACAACCTCCTTGAGTGAGT	
tcf-miR-7	TGGAAGACTAGTGATTTTGTTGTT	
bmo-mir-6497-p5	TCGGGATAAGGATTGGCTC	
tcf-miR-315-5p	TTTTGATTGTTGCTCAGAAGG	
tcf-let-7-5p	TGAGGTAGTAGGTTGTGTGGTT	
tcf-let-7-3p	CTGTACAACTTGCTAACTTTCC	
PC-3p-120711	TGTGGTTGAGCAAAAAGGG	
PC-5p-16060	TCGATCCCCGGCACCTCCA	
PC-3p-74	TGACTAGAGATTCACACTCAT	
tcf-miR-278_R	TCGGTGGGATTCTCGTCCG	
mja-mir-6489-p5	CGGACTGGCGCTCTTGGA	
tcf-miR-10-5p	TACCCTGTAGATCCGAATTTG	
PC-5p-21023	GGTGGAAAGAGATTCAGTCG	
PC-5p-17859	GATGGGTGTGTCTCTGGTGC	
PC-3p-1739	TAGCACCATGTGAATTCAGTAC	
bmo-miR-306a-5p	TCAGGTACTGTGTGACTCTG	
tcf-miR-9a-5p	TCTTTGGTGATCTAGCTGTATG	
tcf-miR-279b	TGACTAGATCCATACTCATCT	
U6-R	AACGCTTCACGAATTTGCGT	

**Table 3 animals-13-01813-t003:** Primer sequences for constructing wild and mutant plasmids of *EsFem-1c* 3′UTR.

Primer	Sequence (5′–3′)	Applications
Fem-1c-miF	GGAGCTCTAACCCTGGAGACAGGGATGAA	Wild plasmids
Fem-1c-miR	CTAGCTAGCCTCGCTGGACAGACAAGATGG	
Fem-1c-248-miF	GGAGCTCGTTGGACAGAGGTTGCTGTTTT	
Fem-1c-248-miR	CTAGCTAGCGGCCCAGAAGTGCTTGATGTT	
Fem-1c-MF1	TGAGGGGGGACTAAATGTCATTTCAATTTTGTGTTGC	Mutant plasmids
Fem-1c-MR1	CATTTAGTCCCCCCTCAAACCTCTCAACAATCACAT	
Fem-1c-MF2	AAAAGTGCCGGGTGTGACGCTGGGGGGGGTCACG	
Fem-1c-MR2	GTCACACCCGGCACTTTTCAAAATTTTTTTTTC	

## Data Availability

The miRNAseq data have been submitted to the BioProject under the accession number PRJNA942040. Upon reasonable request, the datasets used and analyzed during the present study are also available from the corresponding author.

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
