# Peer review of "Eriocheir sinensis feminization-1c (Fem-1c) and Its Predicted miRNAs Involved in Sexual Development and Regulation"

_animals, 2023, doi:10.3390/ani13111813_

Round 1

Reviewer 1 Report

The manuscript entitled “Eriocheir sinensis feminization-1c (Fem-1c) and its predicted miRNAs involved in the sexual development and regulation” by Dandan Zhu, Tianyi Feng, Nan Mo, Rui Han, Wentao Lu and Zhaoxia Cui (animals-2344617) supports that EsFem-1c plays a role in mitten crab AG development and sexual traits development. It is interesting to find that tcf-miR-307 binding to the alternative splicing region with 248 bp (ASR-248) of the 3'UTR sequence with a gender bias could increase expression level, which is helpful to understand the molecular sexual development and regulation mechanism of EsFem-1c in E. sinensis. However, several comments on this manuscript and minor revision of the manuscript are still required.

1. How many duplicate samples have been used for the expression level in the reproduction tissues?

2Line 391: “in vitro” should be italic.

3. In Figure 3, both male and female logos should be adjusted more clearly.

4. In Fig 5a”SR” didn’t explain in figure legend.

4. In Figure 6, please clarify the meaning of “up” and “down”.

5. In the section 2.5, the micro-injection method and injection position should describe more clearly.

6. Line 201-206: There is no space before “×”, the authors need check carefully.

7. The results showed that Fem-1c has alternative splicingbut author didn’t discuss in the discussion.

8. The authors obtain many results, and the discussion seems not enough. I suggested that the discussion can be more detailed that made the research more meaningful.

9. The spelling in references section should be carefully checked to make MS improved.

If the author can improve the language a little more, the manuscript will be more perfect.

Author Response

Dear editor,

Thank you for your letter and the reviewers’ comments concerning our manuscript entitled “Eriocheir sinensis feminization-1c (Fem-1c) and its predicted miRNAs involved in the sexual development and regulation” (ID: animals-2344617). These comments are very helpful for improving our paper. We have read all of the reviewers’ comments carefully and made revisions accordingly. Our revisions and responses to the reviewers are as follows. In the revised manuscript, revisions made according to the comments of Reviewer 1 have been marked as yellow respectively.

-Reviewer 1

  1. How many duplicate samples have been used for the expression level in the reproduction tissues?

For Figure 4, four samples in each group are used for the expression level in the reproduction tissues. We have added the information to the figure legend. Please see line 364 in the revised MS.

  1. 2. Line 391: “in vitro” should be italic.

Done. The word “in vitro” has been changed to be italic. Please see line 428 in the revised MS.

  1. In Figure 3, both male and female logos should be adjusted more clearly.

Done. The male and female logos have been adjusted more clearly. Please see Figure 3 in the revised MS.

  1. In Fig 5a, “SR” didn’t explain in figure legend.

Done. “SR” in the Figure 5a is a typo and has been changed to “SA”. Please see Figure 5a in the revised MS.

  1. In Figure 6, please clarify the meaning of “up” and “down”.

Done. The “up” means that the expression of miRNAs were higher in females than in males, and the “down” means that the expression of miRNAs were higher in males than in females. We have added relevant descriptions to the figure legend. Please see lines 409-410 and 411-412 in the revised MS.

  1. In the section 2.5, the micro-injection method and injection position should describe more clearly.

Done. The dsRNA was injected into the coelom through the arthrodial membrane be-tween the third and fourth pleopod of mature crabs by a 100 μL micro syringe (Shanghai Anting, China). And for juvenile crabs, the injection position was also at the arthrodial membrane between the third and fourth pleopod using the IM11-2 and M-152 micro injection system (NARISHIGE, Japan). We have added these sentences about the method and position of the micro-injection to method. Please see lines 165-166 and 172-174 in the revised MS.

  1. Line 201-206: There is no space before “×”, the authors need check carefully.

Done. We have added the space before “×”. Please see lines 219, 226, and 228 in the revised MS.

  1. The results showed that Fem-1c has alternative splicing, but author didn’t discuss in the discussion.

Done. We have made the relevant discussions about the alternative splicing of EsFem-1c. Pleased see lines 546-548 in the revised MS.

  1. The authors obtain many results, and the discussion seems not enough. I suggested that the discussion can be more detailed that made the research more meaningful.

Done. Besides the discussions about the alternative splicing of EsFem-1c, we also have added some discussions about the differential expression of miRNAs in KEGG pathways and the relationship between gonadal development and tcf-miR-315-5p to make the research more meaningful. Please see lines 515-521 and 528-533 in the revised MS.

  1. The spelling in references section should be carefully checked to make MS improved.

Thanks for your careful review, we have carefully checked the spelling in references section. Please see section references marked with yellow in the revised MS.

Reviewer 2 Report

The manuscript entitled “Eriocheir sinensis feminization-1c (Fem-1c) and its predicted miRNAs involved in the sexual development and regulation” by Dandan Zhu, Tianyi Feng, Nan Mo, Rui Han, Wentao Lu and Zhaoxia Cui (animals-2344617) will be helpful for better understanding of the molecular regulation mechanism of EsFem-1c and provide a resource for future studies of miRNA-mediated regulation of sexual development and regulation in E. sinensis. The results are of interest to the crustacean physiology. Several comments on this manuscript and revision of the manuscript are required.

1. The section 2.1 introduced the mitten crab used in this study, but there is no detailed description of the crab used for RNAi. The authors should provide more information.

2. Please clarify GenBank accession number of EsIAG and EsCFHS-1 in this study.

 3. Please clarify amplification efficiency of primers for qRT-PCR.

 4. Line 78: The species name needs to be italicized.

 5. In the section 2.5, the injection position needs more details.

 6. Line 201-206: There is no space before “×”, the authors need check carefully.

 7. In Figure 2, the asterisk font is different from other fonts, so it needs to be unified.

 8. Line 238: the description with a fusion degree of 80% is too accurate and needs to be adjusted.

 9. How many duplicate samples have been used for each treatment group?

 References: the spelling in references section should be carefully checked to make MS improved.

Minor editing of English language required

Author Response

Dear editor,

Thank you for your letter and the reviewers’ comments concerning our manuscript entitled “Eriocheir sinensis feminization-1c (Fem-1c) and its predicted miRNAs involved in the sexual development and regulation” (ID: animals-2344617). These comments are very helpful for improving our paper. We have read all of the reviewers’ comments carefully and made revisions accordingly. Our revisions and responses to the reviewers are as follows. In the revised manuscript, revisions made according to the comments of Reviewer 2 have been marked as blue respectively.

-Reviewer 2

  1. The section 2.1 introduced the mitten crab used in this study, but there is no detailed description of the crab used for RNAi. The authors should provide more information.

Done. We have provided information on the crab used for RNAi in method section 2.5. The mature mitten crabs have the body weight of 94.27 ± 8.14 g, and the juvenile crabs are on the juvenile Ⅰ stage. Please see lines 160-161 in the revised MS.

  1. Please clarify GenBank accession number of EsIAG and EsCFSH-1 in this study.

Done. We have added GenBank accession number of EsIAG (GenBank ID: KU724192.1) and EsCFSH-1 (GenBank ID: OP351640). Please see lines 169 and 170 in the revised MS.

  1. Please clarify amplification efficiency of primers for qRT-PCR.

Done. The primers EsFem-1c-qF/R had the amplification efficiency of 102.253%, EsCFSH-1-qF/R had the amplification efficiency of 105.895%, EsIAG-qF/R had the amplification efficiency of 101.054%, and the amplification efficiency of Es-β-actin-qF/R was 100.451%. We have added the information to method 2.9. Please see lines 222-225 in the revised MS.

  1. Line 78: The species name needs to be italicized.

Done. The species name has been italicized. Please see line 84 in the revised MS.

  1. In the section 2.5, the injection position needs more details.

Done. The dsRNA was injected into the coelom through the arthrodial membrane be-tween the third and fourth pleopod of mature crabs by a 100 μL micro syringe (Shanghai Anting, China). And for juvenile crabs, the injection position was also at the arthrodial membrane between the third and fourth pleopod using the IM11-2 and M-152 micro injection system (NARISHIGE, Japan). We have added these sentences about the method and position of the micro-injection to method. Please see lines 165-166 (yellow highlight) and 172-174 (yellow highlight) in the revised MS.

  1. Line 201-206: There is no space before “×”, the authors need check carefully.

Done. We have added the space before “×”. Please see lines 219, 226, and 228 in the revised MS.

  1. In Figure 2, the asterisk font is different from other fonts, so it needs to be unified.

Done. We have changed the asterisk font “Times New Roman” to font “Arial” in Figure 2. Please see Figure 2 in the revised MS.

  1. Line 238: the description with a fusion degree of 80% is too accurate and needs to be adjusted.

Done. We have adjusted the description with a fusion degree from “80%” to “70 - 80%”. Please see line 259 in the revised MS.

  1. How many duplicate samples have been used for each treatment group?

Done. For different experiments in our paper, we selected different number of samples to get accurate results. The details are shown at the figure legend of every figure in the revised MS.

The tissue expression profiles of EsFem-1c in normal and intersex mitten crabs is based on four samples (line 319).

The affection analysis of injecting with the dsRNA EsFem-1c is from six samples (lines 343 and 347).

The expression of EsFem-1c in the reproduction tissues during the developmental stages was derived from data from four individuals (line 364).

The relative expression levels of the 16 expressed miRNAs were detected by qRT-PCR in three males and three females (line 415).

The analysis experiments with miRNAs are all performed in three duplicate samples (lines 458 and 484).

Please see in the revised MS.

Reviewer 3 Report

Review report on “Eriocheir sinensis feminization-1c (Fem-1c) and its predicted miRNAs involved in the sexual development and regulation”

A brief summary

The manuscript provides an in-depth study of the role of one gene, fem-1c on sexual development in the Chinese mitten crab. Through silencing of this gene, the authors show that the expression of two other genes (CFSH and IAG) important in crustacean sexual development are altered in different ways. The authors measure the expression of fem-1c during development, finding a down-regulation trend as the reproductive tissues and androgenic gland develop. Then, the authors went on to investigate the microRNAs that may regulate the expression of fem-1c, identifying that there is a difference between male and female crabs in the expression of some of the microRNAs. They identify two miRNAs which they have synthetically made and along with the fem-1c 3’UTR where the miRNAs putatively bind, co-express in mammalian cells to measure luciferase activity. The authors use intersex animals to show that fem-1c is important in the male phenotype – since there was a higher expression of the gene in the androgenic glands of males compared with those from intersex crabs. They conclude from their results that EsFem-1c may be an upstream regulator of IAG and CFSH in this species and describe ways that the miRNAs identified may regulate Fem-1c. The experiments produced very interesting and useful results.

General concept comments

Language let this manuscript down, since there are many areas that were not described clearly, making the experiments and results difficult to understand. Rewording many areas of the manuscript as outlined in the specific comments should rectify this.

Specific comments

Title: Change to “Eriocheir sinensis feminization-1c (Fem-1c) and its predicted miRNAs involved in sexual development and regulation”

Line 12: AG written in full in the summary

Line 20: “cloned” – do you mean you cloned the gene?

Line 48: Reference number 9 is not about crustaceans but locusts, please rectify this.

Line 54: Hyriopsis cumingii is not a crustacean, although you seem to be talking about crustaceans only in this paragraph, so this is misleading.

Line 62 and 63: “considered” and “clarified” are not the correct words to use.

Line 64: “complementarity” is a mistake.

Line 77: “……molecular interactions involved in post-transcriptional regulatory processes…..” would make more sense.

Line 81-82: Which species was this investigation carried out in?

Line 88: Please explain the details of normal and intersexual crabs. Intersex crabs may be a more appropriate term to use.

Line 91-93: This sentence doesn’t make sense – particularly “special miRNA was found to be a gender bias”

Line 93: Also: “….binding to the alternative splicing region of the 3’UTR with 248 bp (ASR-248).” Please clarify “with 248 bp”.

Methods

Line 101: should this read “androgenic glands” not gonads

Line 116: Why did you extract genomic DNA? Please add the reason to the methods.

Table 1 title: remove “are”

Line 155: why do you mention normal salinity just for one group? Were all of the groups kept at the same salinity?

Line 156 and 157: why are the dsRNA called EGFP:EGFP and EsFem-1c:Fem-1c? Why not just call them dsRNA EGFP and dsRNA Fem-1c?

Line 170: Please provide more details of the sample preparation used for the juvenile crabs.

Line 179: I think ploy-N is a typo.

Line 196: What was the reference gene used for the qPCR of the target genes? Is this different to the single-strand U6 primer used as an internal control for the miRNA qPCR? Please add to the methods.

Line 221: It would be useful to have a sentence explaining why the luciferase assay was being carried out. The method does not mention when you applied the miRNA, how long they were incubated with the cells, or the concentrations used. Please add more information to the methods on the use of the miRNAs.

Line 240: When were the cells harvested: what does “harvested for 24 h” mean? Please clarify in the manuscript.

Figure 1: I find it hard to identify the “black blanks” in the figure.

Figure 2: Y-axis label should read “Relative expression level” (please rectify in the other figures). Please use different symbols for each tissue group to show the significant differences – it looks like the I-M sample was not significantly different from the F-O sample because they both have “a”. Alternatively, have different graphs for the three tissue groups. Figure legend: please replace “intersexes” with “intersex”.

Line 291: should be targeted silencing

Line 293: “specific dsRNA EsFem-1c” doesn’t make sense, please reword.

Line 295: Please be more specific about the “crab phenotypes” and next line – should this be gonopores rather than gonophores?

Figure 3: Please define “BC” in the figure legend. Please also define the difference between a and b in the figure legend and also on the figure – “female” and “male”. C: the y-axis label needs to have “the” removed and the units written properly as µm, it is difficult to read.

Line 297: How did you measure the penis length – this should be in the methods please.

Line 304: what do the two percentages refer to – is that in the androgenic gland and the testis respectively? Please add.

Line 318: Please refer to a figure number.

Line 328: testis typo error.

Figure 5: I think that “SR” on the female reproductive system should be “SA”.

Line 346: italicise E. sinensis

Line 352: cluster instead of clusters

Figure 6 Legend: Please state the units of the “norm values derived from sequencing data”.

Figure 7: It is difficult to see the error bars on C – why are there dots there too? Please remove the dots if they don’t show anything. Please alter the “norm” axis to be more specific and include units. To produce Figure 7b, did you use WT fem-1c?

Please change the scale of Figure 7e – it is difficult to see any difference between the two grey bars currently. You could start the y-axis at 0.5 instead of 0.

Line 392: Should this be “While the tcf-miR-7 mimic….” instead of “mimics” – are you referring to bar D on Figure 7b?

Line 395 and lines 400 to 402: should be reworded for clarification.

Line 454: Please insert “male” into the sentence: “higher expression level in normal male AG than in intersex AG……”

Line 455: How is MnFem-1b different to EsFem-1c?

Line 459: should be “knocking down” or “silencing” instead of “knocking”

Line 480: please reword, it doesn’t make sense.

Line 492: Do you mean “gender-biased”?

Review report on “Eriocheir sinensis feminization-1c (Fem-1c) and its predicted miRNAs involved in the sexual development and regulation”

A brief summary

The manuscript provides an in-depth study of the role of one gene, fem-1c on sexual development in the Chinese mitten crab. Through silencing of this gene, the authors show that the expression of two other genes (CFSH and IAG) important in crustacean sexual development are altered in different ways. The authors measure the expression of fem-1c during development, finding a down-regulation trend as the reproductive tissues and androgenic gland develop. Then, the authors went on to investigate the microRNAs that may regulate the expression of fem-1c, identifying that there is a difference between male and female crabs in the expression of some of the microRNAs. They identify two miRNAs which they have synthetically made and along with the fem-1c 3’UTR where the miRNAs putatively bind, co-express in mammalian cells to measure luciferase activity. The authors use intersex animals to show that fem-1c is important in the male phenotype – since there was a higher expression of the gene in the androgenic glands of males compared with those from intersex crabs. They conclude from their results that EsFem-1c may be an upstream regulator of IAG and CFSH in this species and describe ways that the miRNAs identified may regulate Fem-1c. The experiments produced very interesting and useful results.

General concept comments

Language let this manuscript down, since there are many areas that were not described clearly, making the experiments and results difficult to understand. Rewording many areas of the manuscript as outlined in the specific comments should rectify this.

Specific comments

Title: Change to “Eriocheir sinensis feminization-1c (Fem-1c) and its predicted miRNAs involved in sexual development and regulation”

Line 12: AG written in full in the summary

Line 20: “cloned” – do you mean you cloned the gene?

Line 48: Reference number 9 is not about crustaceans but locusts, please rectify this.

Line 54: Hyriopsis cumingii is not a crustacean, although you seem to be talking about crustaceans only in this paragraph, so this is misleading.

Line 62 and 63: “considered” and “clarified” are not the correct words to use.

Line 64: “complementarity” is a mistake.

Line 77: “……molecular interactions involved in post-transcriptional regulatory processes…..” would make more sense.

Line 81-82: Which species was this investigation carried out in?

Line 88: Please explain the details of normal and intersexual crabs. Intersex crabs may be a more appropriate term to use.

Line 91-93: This sentence doesn’t make sense – particularly “special miRNA was found to be a gender bias”

Line 93: Also: “….binding to the alternative splicing region of the 3’UTR with 248 bp (ASR-248).” Please clarify “with 248 bp”.

Methods

Line 101: should this read “androgenic glands” not gonads

Line 116: Why did you extract genomic DNA? Please add the reason to the methods.

Table 1 title: remove “are”

Line 155: why do you mention normal salinity just for one group? Were all of the groups kept at the same salinity?

Line 156 and 157: why are the dsRNA called EGFP:EGFP and EsFem-1c:Fem-1c? Why not just call them dsRNA EGFP and dsRNA Fem-1c?

Line 170: Please provide more details of the sample preparation used for the juvenile crabs.

Line 179: I think ploy-N is a typo.

Line 196: What was the reference gene used for the qPCR of the target genes? Is this different to the single-strand U6 primer used as an internal control for the miRNA qPCR? Please add to the methods.

Line 221: It would be useful to have a sentence explaining why the luciferase assay was being carried out. The method does not mention when you applied the miRNA, how long they were incubated with the cells, or the concentrations used. Please add more information to the methods on the use of the miRNAs.

Line 240: When were the cells harvested: what does “harvested for 24 h” mean? Please clarify in the manuscript.

Figure 1: I find it hard to identify the “black blanks” in the figure.

Figure 2: Y-axis label should read “Relative expression level” (please rectify in the other figures). Please use different symbols for each tissue group to show the significant differences – it looks like the I-M sample was not significantly different from the F-O sample because they both have “a”. Alternatively, have different graphs for the three tissue groups. Figure legend: please replace “intersexes” with “intersex”.

Line 291: should be targeted silencing

Line 293: “specific dsRNA EsFem-1c” doesn’t make sense, please reword.

Line 295: Please be more specific about the “crab phenotypes” and next line – should this be gonopores rather than gonophores?

Figure 3: Please define “BC” in the figure legend. Please also define the difference between a and b in the figure legend and also on the figure – “female” and “male”. C: the y-axis label needs to have “the” removed and the units written properly as µm, it is difficult to read.

Line 297: How did you measure the penis length – this should be in the methods please.

Line 304: what do the two percentages refer to – is that in the androgenic gland and the testis respectively? Please add.

Line 318: Please refer to a figure number.

Line 328: testis typo error.

Figure 5: I think that “SR” on the female reproductive system should be “SA”.

Line 346: italicise E. sinensis

Line 352: cluster instead of clusters

Figure 6 Legend: Please state the units of the “norm values derived from sequencing data”.

Figure 7: It is difficult to see the error bars on C – why are there dots there too? Please remove the dots if they don’t show anything. Please alter the “norm” axis to be more specific and include units. To produce Figure 7b, did you use WT fem-1c?

Please change the scale of Figure 7e – it is difficult to see any difference between the two grey bars currently. You could start the y-axis at 0.5 instead of 0.

Line 392: Should this be “While the tcf-miR-7 mimic….” instead of “mimics” – are you referring to bar D on Figure 7b?

Line 395 and lines 400 to 402: should be reworded for clarification.

Line 454: Please insert “male” into the sentence: “higher expression level in normal male AG than in intersex AG……”

Line 455: How is MnFem-1b different to EsFem-1c?

Line 459: should be “knocking down” or “silencing” instead of “knocking”

Line 480: please reword, it doesn’t make sense.

Line 492: Do you mean “gender-biased”?

Author Response

Dear editor,

Thank you for your letter and the reviewers’ comments concerning our manuscript entitled “Eriocheir sinensis feminization-1c (Fem-1c) and its predicted miRNAs involved in the sexual development and regulation” (ID: animals-2344617). These comments are very helpful for improving our paper. We have read all of the reviewers’ comments carefully and made revisions accordingly. Our revisions and responses to the reviewers are as follows. In the revised manuscript, revisions made according to the comments of Reviewer 3 have been marked as green respectively.

-Reviewer 3

  1. Title: Change to “Eriocheir sinensis feminization-1c (Fem-1c) and its predicted miRNAs involved in sexual development and regulation”

Thanks for your suggestion, we have changed the title to “Eriocheir sinensis feminization-1c (Fem-1c) and its predicted miRNAs involved in sexual development and regulation”.

  1. Line 12: AG written in full in the summary

Done. We have rewritten “AG” to “androgenic gland”. Please see line 12 in the revised MS.

  1. Line 20: “cloned” – do you mean you cloned the gene?

Yes. We have cloned the gene to ensure the accuracy of sequence, especially the 3' UTR sequences.

  1. Line 48: Reference number 9 is not about crustaceans but locusts, please rectify this.

Done. We have deleted reference number 9 to avoid misleading. Please see line 49 in the revised MS.

  1. Line 54: Hyriopsis cumingii is not a crustacean, although you seem to be talking about crustaceans only in this paragraph, so this is misleading.

Done. we have rewritten this sentence to describe exactly what we mean. The sentence “The fem-1c gene has also been identified in some species” to “The fem-1c gene has also been identified in Hyriopsis cumingii and E. sinensis.” Please see line 54 in the revised MS.

  1. Line 62 and 63: “considered” and “clarified” are not the correct words to use.

Done. We have changed two words to “regarded” and “known”. Please see lines 63 and 64 in the revised MS.

  1. Line 64: “complementarity” is a mistake.

Thanks for your careful review, we have rewritten this sentence to describe exactly what we mean. The sentence “MiRNA binds to specific complementarity sequences of mRNA to become active and function by forming the miRNA-induced silencing complex (miRISC)” have been changed to “After forming the miRNA-induced silencing complex (miRISC), miRNA binds to specific mRNA through complementarity sequences to become active and function”. Please see lines 66-68 in the revised MS.

  1. Line 77: “……molecular interactions involved in post-transcriptional regulatory processes…..” would make more sense.

Thanks for your careful review, we have rewritten this sentence to describe exactly what we mean. The sentence “These studies try to explain how miRNAs can target different genes to predict molecular interactions of post-transcriptional regulatory processes” have been changed to “Many studies in E. sinensis have explained how miRNAs control the post-transcriptional regulatory processes”. Please see lines 80-81 in the revised MS.

  1. Line 81-82: Which species was this investigation carried out in?

This investigation was carried out in the Chinese mitten crab Eriocheir sinensis. We have added the species name. Please see lines 84 and 86 in the revised MS.

  1. Line 88: Please explain the details of normal and intersexual crabs. Intersex crabs may be a more appropriate term to use.

Done. The word “intersexual” in this sentence has been changed to “intersex”. Intersex crabs are genetic female crabs with androgenic glands. Please see lines 93-94 in the revised MS.

  1. Line 91-93: This sentence doesn’t make sense – particularly “special miRNA was found to be a gender bias”

Done. we have rewritten this sentence to describe exactly what we mean. This sentence has changed to “it was found that the gender-biased expression pattern of alternative spliced 3'UTR with the length of 248 bp (ASR-248) in EsFem-1c gene was the basis for regulating the function of EsFem-1c” Please see lines 97-100 in the revised MS.

  1. Line 93: Also: “….binding to the alternative splicing region of the 3’UTR with 248 bp (ASR-248).” Please clarify “with 248 bp”.

Done. “With 248 bp” means that the length of alternative splicing region of the 3’UTR is 248 bp. In order to make this sentence easier to understand, we have revised it. This sentence has changed to “…alternative spliced 3'UTR with the length of 248 bp (ASR-248) in EsFem-1c gene…”. Please see line 98 in the revised MS.

  1. Line 101: should this read “androgenic glands” not gonads

Done. we have changed the word to “androgenic glands”. Please see line 109 in the revised MS.

  1. Line 116: Why did you extract genomic DNA? Please add the reason to the methods.

Done. The extracted genomic DNA was used to distinguish the genetic gender of intersex crabs we obtained, by specific sexual primers. To show the usefulness of this method, we have added some descriptions to results. The sentence is “The gender of intersex crabs used in our experiment was distinguished to be female by our specific primers (the data was not shown).”. Please see lines 303-304 in the revised MS.

  1. Table 1 title: remove “are”

Done. We have removed “are” from the title of table 1. Please see Table 1 in the revised MS.

  1. Line 155: why do you mention normal salinity just for one group? Were all of the groups kept at the same salinity?

Done. The “normal salinity” should be “saline solution”. We have replaced this word in the revised MS. Please see line 163 in the revised MS.

  1. Line 156 and 157: why are the dsRNA called EGFP: EGFP and EsFem-1c: Fem-1c? Why not just call them dsRNA EGFP and dsRNA Fem-1c?

Done. We have changed the name of treatment group, the blank control group (saline solution), the negative control group (dsRNA EGFP), and the treated group (dsRNA EsFem-1c). Please see lines 163, 164 and 171, also the Figure 3 in the revised MS.

  1. Line 170: Please provide more details of the sample preparation used for the juvenile crabs.

Done. The sample preparation used for the juvenile crabs is mentioned at the method section 2.5. The juvenile crabs at juvenile Ⅰ stage was collected from the crab hatcheries in Jiangsu province, China. Please see lines 105 and 160-161 in the revised MS. Also, the samples preparation for observation were detailed in method 2.6. The samples for observation were prepared through ethanol gradient dehydration and tert-butyl alcohol replacement. After being frozen for 20 h in -20 ℃, the samples were dried for 24 h in the Martin Christ Alpha 1 – 4LD plus freeze – dryer (Martin Christ, Germany) before being sprayed with gold by a Hitachi E-1010 ion sputtering device (Hitachi, Japan). Please see lines 181-185 in the revised MS.

  1. Line 179: I think ploy-N is a typo.

Yes, we have changed the word “ploy-N” to “ambiguous nucleotides”. Please see lines 195-196 in the revised MS.

  1. Line 196: What was the reference gene used for the qPCR of the target genes? Is this different to the single-strand U6 primer used as an internal control for the miRNA qPCR? Please add to the methods.

Done. The single-strand U6 primer was used as an internal control for the miRNA qPCR, while Esβ-actin gene (ATO74508.1) was used as the internal control for the mRNA qRT-PCR. They both could ensure the standardization of expression, but they are different in the detecting objects. We have added the description of Esβ-actin to the method section 2.9. The sentence is “To standardize the mRNA expression, Es-β-actin (GenBank ID: ATO74508.1) was used as the reference gene.”. Please see lines 220-221 in the revised MS.

  1. Line 221: It would be useful to have a sentence explaining why the luciferase assay was being carried out. The method does not mention when you applied the miRNA, how long they were incubated with the cells, or the concentrations used. Please add more information to the methods on the use of the miRNAs.

Done. We have added the information to method section to make the assay more clearly.

The sentence explaining why the luciferase assay was being carried out is “To determine whether the miRNAs can regulate EsFem-1c gene, the interactions between them were examined using dual‐luciferase reporter assay in vitro.”. Please see lines 242-243 in the revised MS.

The method about the details of transfection in cells also have been complemented. The sentences include “When the cells were about 70 – 80% confluent, the co-transfection was conducted with 30 ng miRNAs mimics or the negative control (NC), 40 ng pGL4.74 [hRluc/TK] plasmid, and 200 ng Fem1c-WT or Fem1c-MUT plasmid, using Lipofectamine 3000 reagent (Thermo Fisher Scientific, China) according to the manufacturer's protocol.” and “Cells were incubated in serum- and antibiotic-free DMEM for 6 h, then being 48 – h transfection.”. Please see lines 259-262 and 265-266 in the revised MS.

  1. Line 240: When were the cells harvested: what does “harvested for 24 h” mean? Please clarify in the manuscript.

Done. There is a mistake about the cell harvesting. We have rewritten this sentence to express clearly. The sentence is “After harvesting cell lysates,”. Please see line 268 in the revised MS.

  1. Figure 1: I find it hard to identify the “black blanks” in the figure.

Done. We have changed the color of the black blanks to green blanks, and the yellow highlight to gray highlight in the Figure 1 for clarity. Please see Figure 1 and its legend (lines 293 and 296) in the revised MS.

  1. Figure 2: Y-axis label should read “Relative expression level” (please rectify in the other figures). Please use different symbols for each tissue group to show the significant differences – it looks like the I-M sample was not significantly different from the F-O sample because they both have “a”. Alternatively, have different graphs for the three tissue groups. Figure legend: please replace “intersexes” with “intersex”.

Thanks for your careful review. The figure has been optimized as you suggested. Please see Figure 2 and its legend (lines 316-318) in the revised MS.

  1. Line 291: should be targeted silencing

Done. We have corrected the word. Please see line 321 in the revised MS.

  1. Line 293: “specific dsRNA EsFem-1c” doesn’t make sense, please reword.

Done. The word “specific” has been deleted. Please see line 323 in the revised MS.

  1. Line 295: Please be more specific about the “crab phenotypes” and next line – should this be gonopores rather than gonophores?

Done. We have added the description about the crab phenotypes, including “, especially gonopore's cover in female crab and AG in male crab,”. And the word “gonophores” have changed to “gonopores”. Please see lines 325-326 and 327 in the revised MS.

  1. Figure 3: Please define “BC” in the figure legend. Please also define the difference between a and b in the figure legend and also on the figure – “female” and “male”. C: the y-axis label needs to have “the” removed and the units written properly as µm, it is difficult to read.

Done. The figure has been optimized as you suggested. First, “BC: blank control group” has been added to lines 341-342 at the figure legend in the revised MS. Second, the difference between a and b in the figure legend and also on the figure represented the significant difference between treated groups of each gene. We have replenished the definition to the figure legend. Please see lines 347-348 at Figure 3 legend in the revised MS.

  1. Line 297: How did you measure the penis length – this should be in the methods please.

Done. We measured the penis length by the software Image J, and converted to get the data according to the scale of scanning electron microscope picture. The sentence of “The length of male penis was obtained by the software Image J and converted according to the scale of SEM images.” has been added to method 2.6. Please see lines 185-186 in the revised MS.

  1. Line 304: what do the two percentages refer to – is that in the androgenic gland and the testis respectively? Please add.

Done. The information of the two percentages have been added with the sentence of “…in AG and testis   ”. Please see line 335 in the revised MS.

  1. Line 318: Please refer to a figure number.

Done. The figure number has been referred. Please see lines 352 and 355 in the revised MS.

  1. Line 328: testis typo error.

Done. We have changed the word to “testis”. Please see line 361 in the revised MS.

  1. Figure 5: I think that “SR” on the female reproductive system should be “SA”.

Done. “SR” in the Figure 5a is a typo and has been changed to “SA”. Please see Figure 5a in the revised MS.

  1. Line 346: italicise E. sinensis

Done. We have italicized the word “E. sinensis”. Please see line 379 in the revised MS.

  1. Line 352: cluster instead of clusters

Done. We have changed the word to “cluster”. Please see line 385 in the revised MS.

  1. Figure 6 Legend: Please state the units of the “norm values derived from sequencing data”.

Done. The “norm values derived from sequencing data” do not have units. The norm values are normalized from miRNA read counts to count the expression of miRNA in different samples. Therefore, in the method 2.8, the sentence of “The miRNA read counts were normalized to norm values to be a measure of miRNA expression,” have added, and the sentence of “The norm values are normalized from miRNA read counts to count the expression of miRNA.” have been added to figure 6 legend. Please see lines 203-204 and 413-414 in the revised MS.

  1. Figure 7: It is difficult to see the error bars on C – why are there dots there too? Please remove the dots if they don’t show anything. Please alter the “norm” axis to be more specific and include units. To produce Figure 7b, did you use WT fem-1c?

Done. The dots on the column of figure 7c represent the data used for analysis. They show the distribution of data intuitively and clearly, and enhance the credibility of the chart. As we answered in Question 36, the norm values do not have units, but we have changed the “norm” axis to “norm value” axis to make the description more accurate. And in figure 7b, the plasmid WT fem-1c was used to get the figure. The X-axis of figure 7b also has been optimized for easy understanding. Please see Figure 7 in the revised MS.

  1. Please change the scale of Figure 7e – it is difficult to see any difference between the two grey bars currently. You could start the y-axis at 0.5 instead of 0.

Done. The figure 7e has been optimized as suggested. The Y-axis started at 0.5. Please see Figure 7e in the revised MS.

  1. Line 392: Should this be “While the tcf-miR-7 mimic….” instead of “mimics” – are you referring to bar D on Figure 7b?

Yes. We have changed “the tcf-miR-7 mimics” to “the tcf-miR-7 mimic”. Please see line 429 in the revised MS.

  1. Line 395 and lines 400 to 402: should be reworded for clarification.

Done. We have rewritten these sentences for clarification.

The sentence “The miRNA tcf-miR-315-5p was selected for further verification based on the maximum reduction.” have been replaced by “The miRNA tcf-miR-315-5p was selected for further analysis and verification, because it had the greatest effect on reducing luciferase activity.”. Please see lines 433-434 in the revised MS.

And the sentence “The results of co-transfecting with Fem1c-MUT plasmid and tcf-miR-315-5p mimics indicated that the luciferase activity decreased by 6.5% (p < 0.05). In contrast, co-transfecting with Fem1c-WT and tcf-miR-315-5p mimics decreased by 13.8% (p < 0.05), which differed between them (p < 0.05) (Figure 7e).” have been replaced by “In figure 7e, the results of co-transfecting showed that the fluorescence intensity of cells treated with Fem1c-WT plasmid and tcf-miR-315-5p mimic was significantly decreased compared with that of controls (p < 0.05), indicating that tcf-miR-315-5p inhibited the expression of EsFem-1c gene by targeting its 3'UTR. Co-transfecting with Fem1c-MUT plasmid and tcf-miR-315-5p mimic also significantly decreased the luciferase activity (p < 0.05). However, the luciferase activity in Fem1c-MUT plasmid group was higher than that in Fem1c-WT plasmid group (p < 0.05) (Figure 7e).”. Please see lines 438-445 in the revised MS.

  1. Line 454: Please insert “male” into the sentence: “higher expression level in normal male AG than in intersex AG……”

Done. We have inserted “male” into this sentence. Please see line 500 in the revised MS.

  1. Line 455: How is MnFem-1b different to EsFem-1c?

Firstly, MnFem-1b is Fem-1b gene in oriental river prawn Macrobrachium nipponense, and EsFem-1c is Fem-1c gene in the Chinese mitten crab Eriocheir sinensis.

Secondly, according to the analysis of secondary structure and conserved structure domain, MnFem-1b and EsFem-1c both belong to the Fem – 2 family, while they belong to different subfamilies respectively.

Then, MnFem1b shows the highest expression level in testis, while the highest expression of EsFem-1c is in muscle. Furthermore, the expression of Mnfem1b in male testis is higher than that in female ovary, while Esfem-1c in female ovary is higher than that in male testis. Finally, the expression level of MnFem1b is high in the late stage of larval development, while EsFem-1c expresses highly in the early stage of embryonic development.

All in all, MnFem-1b is indeed different to EsFem-1c.

  1. Line 459: should be “knocking down” or “silencing” instead of “knocking”

Done. The word “knocking” has been changed to “knocking down”. Please see line 505 in the revised MS.

  1. Line 480: please reword, it doesn’t make sense.

Done. We have rewritten this sentence to “Considering to the various roles of NAT, the ASR-248 might influence the function of EsFem-1c by changing mRNA stability, masking miRNA-binding sites, and formation of endogenous siRNAs”. Please see lines 537-539 in the revised MS.

  1. Line 492: Do you mean “gender-biased”?

Yes. We have changed the word to “gender-biased”. Please see line 555 in the revised MS.